# Asymmetries around the visual field: From retina to cortex to behavior

**Eline R. Kupers**[1,2¤a]*, **Noah C. Benson**[1,2¤b], **Marisa Carrasco**[1,2], **Jonathan Winawer**[1,2]

**1** Department of Psychology, New York University, New York, New York, United States of America, **2** Center for Neural Sciences, New York University, New York, New York, United States of America

¤a Current address: Department of Psychology, Stanford University, Stanford, California, United States of America
¤b Current address: eScience Institute, University of Washington, Seattle, Washington, United States of America
* eline.kupers@nyu.edu

**Data Availability Statement:** Both simulation and analyses are publicly available via Github: https://github.com/elinekupers/pf_RV1/. Data structures created by the simulation and analyses are

## Abstract

Visual performance varies around the visual field. It is best near the fovea compared to the periphery, and at iso-eccentric locations it is best on the horizontal, intermediate on the lower, and poorest on the upper meridian. The fovea-to-periphery performance decline is linked to the decreases in cone density, retinal ganglion cell (RGC) density, and V1 cortical magnification factor (CMF) as eccentricity increases. The origins of polar angle asymmetries are not well understood. Optical quality and cone density vary across the retina, but recent computational modeling has shown that these factors can only account for a small percentage of behavior. Here, we investigate how visual processing beyond the cone photon absorptions contributes to polar angle asymmetries in performance. First, we quantify the extent of asymmetries in cone density, midget RGC density, and V1 CMF. We find that both polar angle asymmetries and eccentricity gradients increase from cones to mRGCs, and from mRGCs to cortex. Second, we extend our previously published computational observer model to quantify the contribution of phototransduction by the cones and spatial filtering by mRGCs to behavioral asymmetries. Starting with photons emitted by a visual display, the model simulates the effect of human optics, cone isomerizations, phototransduction, and mRGC spatial filtering. The model performs a forced choice orientation discrimination task on mRGC responses using a linear support vector machine classifier. The model shows that asymmetries in a decision maker's performance across polar angle are greater when assessing the photocurrents than when assessing isomerizations and are greater still when assessing mRGC signals. Nonetheless, the polar angle asymmetries of the mRGC outputs are still considerably smaller than those observed from human performance. We conclude that cone isomerizations, phototransduction, and the spatial filtering properties of mRGCs contribute to polar angle performance differences, but that a full account of these differences will entail additional contribution from cortical representations.

permanently archived on the Open Science
Framework URL: https://osf.io/ywu5v/.

**Funding:** This research was supported by the US
NIH R01-EY027401 (M.C. and J.W.). The funders
had no role in study design, data collection and
analysis, decision to publish, or preparation of the
manuscript.

**Competing interests:** The authors have declared
that no competing interests exist.

## Author summary

The neural circuitry of the visual system is organized into multiple maps of the visual
field. Each map is orderly, as nearby cells represent nearby points in the visual field. Each
map is also non-uniform, in that some portions of the visual field are sampled more
densely than others. These non-uniformities emerge from the first stage of processing, the
photoreceptor array in the retina. The cone photoreceptors vary in density with eccentric-
ity—they are denser in the central than the peripheral retina—and with polar angle—they
are denser on the horizontal than vertical meridian. Our analyses show that both the
eccentricity gradient and polar angle asymmetries become more pronounced in each of
two subsequent processing stages, the midget retinal ganglion cells and primary visual
cortex. We then implement a computational observer model incorporating several com-
ponents of the early visual system. The model shows that the information present in the
cone array can explain a small portion of the polar angle asymmetries in human visual
performance, and the information present in the midget retinal ganglion cells can explain
more, but still less than half, of the performance asymmetries. A full account of perfor-
mance asymmetries will entail additional contributions from cortex.

## Introduction

Visual performance is not uniform across the visual field. The most well-known effect is a
decrease in visual acuity as a function of eccentricity: we see more poorly in the periphery
compared to the center of gaze [1–4]. This observed difference in visual performance has been
attributed to several physiological factors, starting as early as the distribution of photoreceptors
[5,6]. In the human fovea, the cones are tightly packed such that visual input is encoded at
high spatial resolution. In peripheral retinal locations, cones are larger and interspersed
among rods, resulting in a drastically lower density [7–10]; hence a decrease in spatial
resolution.

Visual performance also differs as a function of polar angle. At matched eccentricity, per-
formance is better along the horizontal than vertical visual meridian (horizontal-vertical
anisotropy or "HVA", *e.g.*, [11–16]) and better along the lower than upper vertical visual
meridian (vertical-meridian asymmetry or "VMA", *e.g.*, [12–18]). These polar angle asymme-
tries are observed in many different visual tasks, such as those mediated by contrast sensitivity
[12–15,19–31], spatial resolution [11,16,17,19,20,32–34], contrast appearance [35], visual
search [36–44], crowding [44–47], and tasks that are thought to recruit higher visual areas
such as visual working memory [34]. Covert spatial attention improves performance similarly
at all iso-eccentric stimulus locations, thus it does not eliminate the polar angle asymmetries
[12,13,48,49].

These polar angle effects can be large. For instance, for a Gabor patch at 4.5˚ eccentricity
with a spatial frequency of 4 cycles per degree, contrast thresholds are close to double for the
upper vertical meridian compared to the horizontal meridian [12,13,15]. This is an effect size
similar to doubling stimulus' eccentricity from 4.5˚ to 9˚ on the horizontal axis [15,20]. Addi-
tionally, these performance differences are retinotopic, shifting in line with the retinal location
of the stimulus rather than its location in space [14].

The visual system has polar angle asymmetries from its earliest stages, including in the
optics and cone density. In a computational observer model that tracked information from the
photons in the scene through the optics and cone isomerizations, variations in optical quality
and cone density accounted for less than 10% of the observed polar angle asymmetries in a

contrast threshold task [50]. This leads to the question, what additional factors later in the visual processing stream give rise to visual performance differences with polar angle?

One possibility is that even without additional asymmetries in cell density, later processing could exacerbate the earlier asymmetries. For example, the larger cone apertures observed at lower cone densities result in greater downregulation of the cone photocurrent [51], hence this decrease in signal-to-noise ratio might exacerbate polar angle asymmetries.

A second—not mutually exclusive—possibility is that there are additional polar angle asymmetries in the distribution of other downstream cell types. In the human retina, the best described retinal ganglion cells (RGCs) are the midget and parasol cells. Both of these cell types show a decrease in density as a function of eccentricity and vary in density as a function polar angle in humans [52–58] and monkeys [59–62]. Because midget RGCs are the most numerous ganglion cells in primates (*i.e.*, 80% of ~1 million RGCs compared to 10% parasols and 10% other types) and have small cell bodies and small dendritic field trees that increase with eccentricity [60,61,63], they are often hypothesized to set an anatomical limit on high resolution spatial vision such as acuity and contrast sensitivity at mid to high spatial frequencies [55,61].

Interestingly, in the range of eccentricities used for many psychophysical tasks (0–10˚), cone density shows an HVA (greater density on the horizontal than vertical meridian), but not a VMA, inconsistent with behavior (there is a slightly greater density on the upper than lower vertical visual meridian, opposite what one would predict to explain behavior) [8–10]. Midget RGC density, in contrast, shows both an HVA and a VMA, making their distribution patterns more similar to behavioral patterns [52–54,57,64].

Here, we investigate how asymmetries in the visual system vary across processing stages. First, we quantify asymmetries in spatial sampling around the visual field in three early visual processing stages: cones, mRGCs, and V1 cortex. We do so because it is important to first identify *if* there are any differences in spatial encoding across these processing stages, and if so, *how* these differences relate to differences in behavior. Then we extend our previously published computational observer model, which included optics and cone sampling, by adding a model of conversion from photon absorptions to photocurrent, and then mRGC-like spatial filtering. We compare this observer model to our previous model (no RGC layer) and to human performance on a two alternative forced choice (2-AFC) orientation discrimination task. By comparing the predicted performance to human observers, we can quantify the contribution of mRGCs to visual performance differences around the visual field.

## Results

We quantify the asymmetries in cone density, midget retinal ganglion cells (mRGCs) density and V1 cortical magnification factor (CMF)—both as a function of eccentricity and for the four cardinal meridians. In the next two sections, we first show that both eccentricity gradients and polar angle asymmetries are amplified from cones to mRGCs and from mRGCs to early visual cortex. Then we implement the observed variations in mRGC density in a computational observer model to test whether biologically plausible differences in mRGC sampling across the cardinal meridians can quantitatively explain psychophysical performance differences as a function of polar angle.

### Fovea-to-periphery gradient is amplified from retina to mRGCs to early visual cortex

A hallmark of the human retina is the sharp drop in cone density from fovea to periphery [8–10]. Within the central one degree, cone density decreases dramatically (on average by

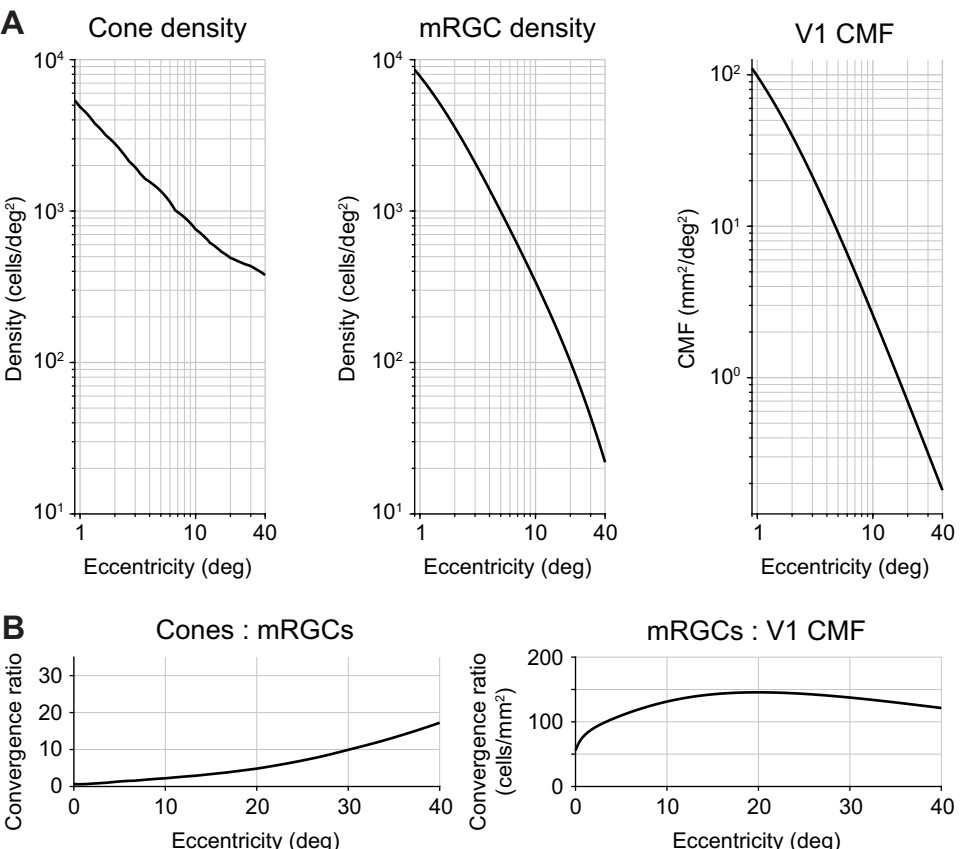

**Fig 1. Foveal over-representation is amplified from cones to mRGCs to cortex.** (A) Cone density, mRGC receptive field density, and V1 cortical magnification factor as a function of eccentricity. Left panel: Cone data from Curcio *et al.* [9]. Middle panel: midget RGC RF density data from Watson [64]. Both cone and mRGC data are the average across cardinal retinal meridians of the left eye using the publicly available toolbox ISETBIO [65–67]. Right panel: V1 CMF is predicted by the areal equation published in Horton and Hoyt [68]. (B) Transformation ratios from cones to mRGCs and mRGCs to V1. The cone:mRGC ratio is unitless, as both cone density and mRGC density are quantified in cells/deg$^2$. The increasing ratio indicates higher convergence of cone signals by the mRGCs. For mRGC:V1 CMF ratio units are defined in cells/mm$^2$. The ratio increase in the first 20˚ indicates an amplification of the foveal over-representation in V1 compared to mRGCs.

3.5-fold). Beyond the fovea, cone density continues to decrease by 10-fold between 1˚ and 20˚ eccentricity (**Fig 1A**, left panel). This decrease in cone density is due to an increase in cone spacing caused by the presence of rods and by the increase in cone diameter [9].

The second processing stage we focus on are the midget RGCs. The mRGC cell bodies are laterally displaced from their receptive fields by the foveal cones. Therefore, we use a computational model by Watson [64] that combines cone density [9], mRGC density [53] and displacement [57] to infer the mRGC density referred to the visual field, rather than the cell body positions. Throughout, we refer to mRGC density with respect to receptive fields. Like the cones, midget RGCs sample the visual field differentially as a function of eccentricity. At the central one degree, mRGC density is greater than cone density. The fovea-to-periphery gradient is steeper for mRGCs than for cones (**Fig 1A,** middle panel compared to left panel). This *divergence* results in a cone:mRGC ratio of 0.5 (**Fig 1B**, left panel), indicating a 'direct line' between a single cone and a pair of ON- and OFF-center mRGCs. In the periphery, mRGC density falls off at a faster rate than cones. For example, cone density decreases by 10-fold between 1˚ and 20˚ eccentricity, whereas mRGC density decreases by 80-fold. This *convergence*

can also be expressed in the cone:mRGC ratio, which increases as a function of eccentricity (**Fig 1B**, left panel).

Third, we quantify the amount of V1 surface area devoted to a portion of the visual field, also known as the cortical magnification factor (**Fig 1A**, right panel). There have been claims that V1 CMF is proportional to retinal ganglion cell density [69–72] and see Discussion). However, when comparing human mRGCs density [64] to V1 CMF [68], we find that the ratio is not constant: The foveal magnification is even more accentuated in V1 up to 20˚ eccentricity (**Fig 1B**, right panel). These results are consistent with the findings in squirrel monkey [73], owl monkey [74], and macaque [75], all of which show that the cortical magnification function falls off with eccentricity more steeply in V1 than would be predicted by mRGC density alone. Beyond 20˚ eccentricity, the mRGC to V1 CMF ratio declines slowly. This effect is driven by V1 CMF falling off slightly more steeply than mRGC density. The relative compression of V1 CMF vs mRGC density in the far periphery has been reported in owl monkey [74]. However, given that this result has not been confirmed in human cortex, we cannot exclude the possibilities that in the far periphery Watson's formula [64] overpredicts mRGC density, Horton and Hoyt's formula [68] underpredicts V1 CMF, or a combination of both.

## Polar angle asymmetries are amplified from cones to mRGCs

Cone density differs as a function of polar angle. It is higher along the horizontal visual field meridian (average of nasal and temporal retina meridians) than the upper and lower vertical visual field meridians (representing the inferior and superior retinal meridians) (**Fig 2A**, left panel). This horizontal-vertical asymmetry is around 20% and relatively constant with eccentricity. There is no systematic difference between the cone density in the upper and lower visual field meridians. If anything, there is a slight 'inverted' vertical-meridian asymmetry in the central three degrees: cones are more densely packed along the upper vertical visual meridian. Assuming greater density leads to better performance, this would predict better performance on the upper vertical meridian in the central three degrees, opposite of the typical asymmetry reported in behavior, which has been found up to 1.5˚ eccentricity in a study on contrast sensitivity [30]. All of these patterns of cone density asymmetries are found using two different datasets with different methods: a post-mortem retinal dataset [9] and an *in vivo* dataset [10], indicating reproducibility of the biological finding. All of the patterns are also consistent when computed using two different analysis toolboxes (ISETBIO [65–67] and rgcDisplacementMap [76], **S1 Fig**, top row), indicating computational reproducibility.

The polar angle asymmetries in density are larger in the mRGC distribution. The horizontal visual field meridian (average of nasal and temporal retina) contains higher cell densities (after correction for cell body displacement) than the upper and lower visual field meridians (**Fig 2A**, middle panel). This horizontal-vertical asymmetry increases with eccentricity. For example, at 3.5˚ eccentricity, the average horizontal visual field density is ~20% higher than the average of upper and lower visual field meridians. By 40˚ eccentricity, this density difference increases to ~60%. Beyond 10˚ eccentricity, this horizontal-vertical asymmetry is mostly driven by the nasal retina, as it contains higher mRGC density than the temporal retina. This finding is in line with earlier histology reports in macaque [62] and positively correlated with spatial resolution tasks (*e.g.*, [77]). This nasal-temporal asymmetry, although interesting, is beyond the focus of this paper, as the asymmetries in performance we observe are found in both binocular and monocular experiments [12,16]. Overall, the emphasis on the horizontal is substantially greater in the mRGCs than the cones.

Unlike the cones, mRGC receptive fields show a consistent asymmetry along the vertical meridian: The lower visual meridian (superior retinal meridian) contains a higher mRGC

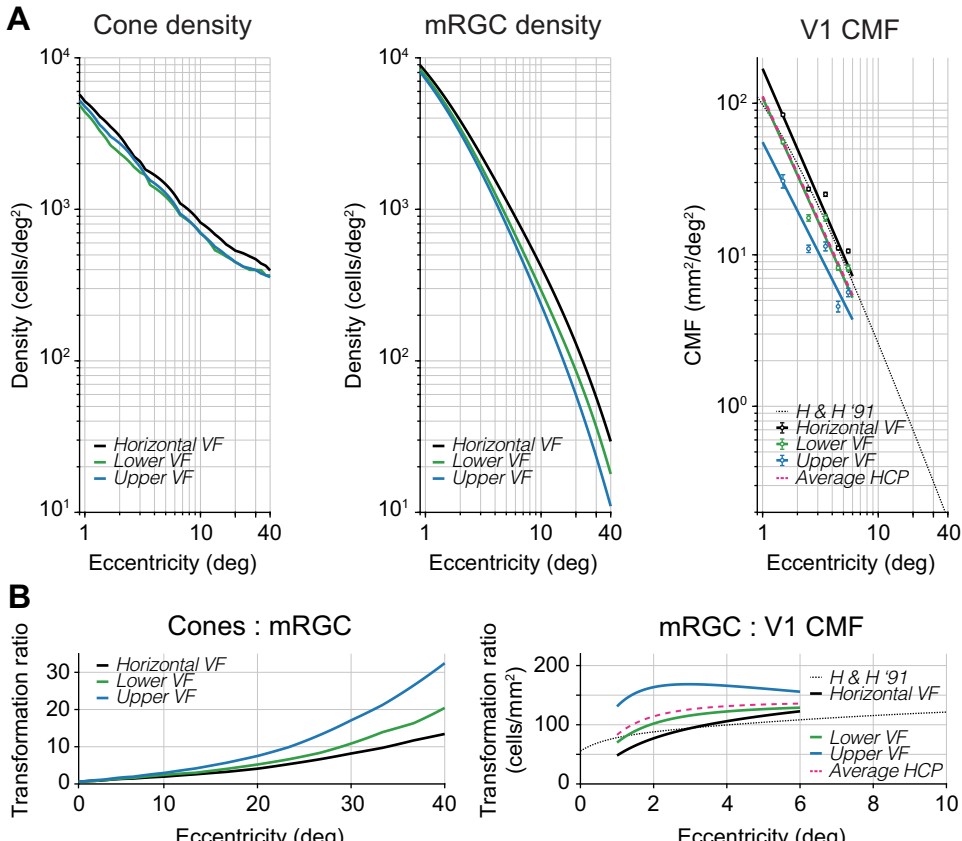

**Fig 2. Nonuniformities in polar angle representations are amplified from cones to mRGCs to cortex.** (A) Cone density, mRGC density, and V1 CMF for cardinal meridians as a function of eccentricity. Left panel: Cone density from Curcio *et al.* [9]. Middle panel: mRGC densities from Watson [64]. All data are in visual field coordinates. Black line represents the horizontal visual field meridian (average of nasal and temporal retina), green line represents lower visual field meridian (superior retina), and blue line represents upper visual field meridian (inferior retina). Cone and mRGC data are computed with the open-source software ISETBIO [65–67]. Right panel: V1 CMF computed from the HCP 7T retinotopy dataset analyzed by Benson *et al.* [78] (black, green, blue dots and lines) and predicted areal CMF by the formula in Horton and Hoyt [68] (dotted black line, replotted from Fig 1). All data are plotted in visual field coordinates where black, green, and blue data points represent the horizontal, lower, and upper visual field meridians, respectively. Data points represent the median V1 CMF of ±20˚ wedge ROIs along the meridians for 1–6˚ eccentricity in 1˚ bins. Error bars represent 68%-confidence intervals across 163 subjects using 1,000 bootstraps. Black, green, and blue lines are 1/eccentricity power functions fitted to corresponding data points. Pink dashed line is the average of fits to horizontal, upper, and lower visual field meridians from HCP 7T retinotopy dataset [78] and agrees well with Horton and Hoyt's formula [68]. (B) Transformation ratios from cones to mRGCs and mRGCs to V1 CMF. Ratios are shown separately for the horizontal (black), lower (green) and upper (blue) visual field meridians. The mRGC:V1 CMF panel has a truncated x-axis due to the limited field-of-view during cortical measurements. These polar angle asymmetries can be found across two different computational models of mRGC density (see **S1 Fig**, second row).

density than the upper visual meridian (inferior retinal meridian). This is consistent with the psychophysical VMA, showing better performance on the lower vertical meridian [12–15,19–31]. This asymmetry increases with eccentricity. For example, the lower vertical meridian (superior retina) has ~15% higher density compared to upper vertical (inferior) at 3.5˚, and ~50% higher density at 40˚ eccentricity. This interaction between retinal meridian and eccentricity is summarized in the cone-to-mRGC transformation plot (**Fig 2B**, left panel), where the convergence ratio from cones to mRGCs increases more rapidly along the upper than the lower vertical and the horizontal visual meridians (see also **S2 Fig**).

## Polar angle asymmetries are amplified from mRGCs to early visual cortex

Because the areal V1 CMF calculation by Horton and Hoyt [68] does not make separate predictions for the cardinal meridians, we used the publicly available retinotopy dataset from the Human Connectome Project (HCP) analyzed by Benson *et al.* [79] to calculate the CMF along the meridians (see also [78]). As a first check on agreement between the two datasets, we found that the V1 CMF data measured in 163 subjects with functional MRI [78], pooled across all polar angles, was a close match to Horton and Hoyt's [68] prediction based on lesion case studies from three decades ago. We then used the HCP dataset to compute CMF along the separate meridians.

We find that polar angle asymmetries in cortical magnification factors are yet larger than those found in mRGC density (**Fig 2A**, right panel), where V1 CMF is higher on the horizontal than vertical meridian, and the V1 CMF is higher for the lower than the upper vertical meridian. For example, at 3.5˚ eccentricity CMF is ~52% higher on the horizontal than vertical meridians and ~41% higher for the lower than upper vertical meridian. These polar angle asymmetries show a 2x increase within the first three degrees of eccentricity before flattening (**Fig 2B**, right panel) and are mostly driven by the upper vertical meridian (**S2 Fig**). This indicates that the mapping of the visual field in early visual cortex is not simply predicted from the distribution of midget retinal ganglion cells, but rather the cortex increases the retinal polar angle asymmetries.

## A computational observer model from stimulus to mRGCs to behavior

To understand how polar angle asymmetries in visual field representations might affect visual performance, we added a photocurrent transduction and retinal ganglion cell layer to our computational observer model [50]. In this observer model, we used the publicly available ISETBIO toolbox [65–67] to simulate the first stages of visual pathway including the stimulus scene, fixational eye movements, chromatic and achromatic optical aberrations, and isomerization by the cone array. Combining model output with a linear support vector machine classifier allowed us to simulate performance on a 2-AFC orientation discrimination task given information available in the cones. When matching stimulus parameters in the model to a previously published psychophysical experiment [13], we showed that biologically plausibly variations in optical quality and cone density together would contributed no more than ~10% to the observed polar angle asymmetries in contrast sensitivity.

Given the inability of cone density to quantitatively explain differences in visual performance, we extended our model further into the retina to include temporal and spatial filtering, and noise at two later processing stages. First, we added temporal filtering and noise in the conversion of cone isomerizations to photocurrent in the cone outer segments. Second, we added spatial filtering and noise in a model of midget RGCs. The mRGCs are especially interesting because they show a systematic asymmetry between the upper and lower visual field (where the cones did not), and an amplification of the horizontal-vertical asymmetry. The mRGC computational stage is implemented after cone isomerizations and photocurrent and before the model performs the discrimination task. We provide a short overview of the modeled stages that precede the mRGC layer, as details of these stages can be found in our previous paper [50], followed by a discussion of the implementation details of the photocurrent transduction and mRGC layer.

**Scene radiance.** The first stage of the model comprises the photons emitted by a visual display. This results in a time-varying scene defined by the spectral radiance of an achromatic low contrast Gabor stimulus (**Fig 3**, panel 1). The Gabor was oriented 15˚ clockwise or counter-clockwise from vertical with a spatial frequency of 4 cycles per degree. These stimulus

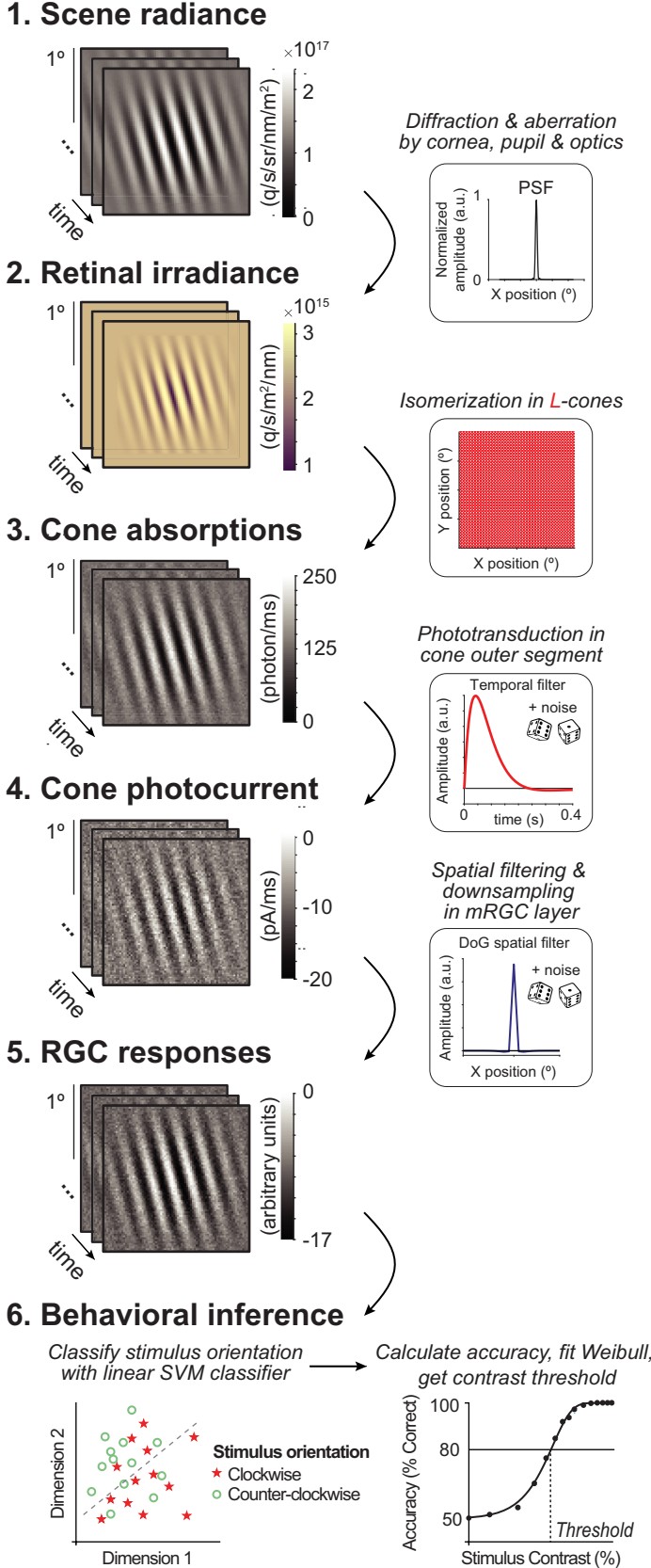

**Fig 3. Overview of computational observer model with additional mRGC layer.** A 1-ms frame of a 100% contrast Gabor stimulus is used at each computational step for illustration purposes. (1) Scene radiance. Photons emitted by the visual display, resulting in a time-varying scene spectral radiance. Gabor stimulus shows radiance summed across 400–700 nm wavelengths. (2) Retinal irradiance. Emitted photons pass through simulated human cornea, pupil, and optics, indicated by the schematic point spread function (PSF) in the top right-side box, resulting in time-varying retinal irradiance. Gabor stimulus shows irradiance with wavelengths converted to RGB values for illustration purposes. (3) Cone absorptions. Retinal irradiance is isomerized by a rectangular cone mosaic, resulting in time-varying photon absorption rates for each L-cone with Poisson noise. (4) Cone photocurrent. Absorptions are converted to photocurrent via temporal integration, gain control, followed by adding Gaussian white noise. This results in time-varying photocurrent for each cone. (5) Midget RGC responses. Time-varying cone photocurrents are convolved with a 2D Difference of Gaussians spatial filter (DoG), followed by additive Gaussian white noise and subsampling. (6) Behavioral inference. A linear support vector machine (SVM) classifier is trained on the RGC outputs to classify stimulus orientation per contrast level. With 10-fold cross-validation, left-out data are tested, and accuracy is fitted with a Weibull function to extract the contrast threshold at ~80%.

parameters were chosen to match a recent psychophysical experiment [15] to later compare model and human performance.

**Retinal irradiance.** The second stage simulates the effect of emitted photons passing through the human cornea, pupil, and lens. This computational step results in time-varying retinal irradiance (**Fig 3**, panel 2). Optics are modeled as a typical human wavefront with a 3-mm diameter pupil without defocus and contain a spectral filter that reduces the fraction of short wavelengths (due to selective absorption by the lens). We do not vary the optics across the different simulations.

**Cone absorptions.** The third stage implements a rectangular cone mosaic with L-cones only (2x2° field-of-view). For each cone, we compute the number of photons absorbed in each 2-ms bin, resulting in a 2D time-varying cone absorption image (**Fig 3**, panel 3). The number of absorptions depends on the photoreceptor efficiency, on the wavelengths of light, and on Poisson sampling due to the quantal nature of light. This stage differs in two ways from our previous model. First, we use an L-cone only retina, and second, we exclude fixational eye movements. We make these two simplifications to keep the model tractable and the calculations to reasonable size. As we describe in the Methods, the number of trials is much larger than in our previous work (to ensure that the classifier has sufficient information to learn the best classification), the number of conditions simulated is much larger (because we vary both cone density and mRGC:cone ratios), and the noise level is substantially higher (because we add noise at phototransduction and mRGC stages). The lack of eye movements enables us to average time points across trials, greatly speeding up processing, as well as simplifying the interpretation of how the new stages contributed to performance.

**Cone photocurrent.** The fourth stage converts photon absorptions to photocurrent, incorporating the recently added phototransduction functionality in ISETBIO by Cottaris *et al.* [51], Here, phototransduction is implemented as a temporal filter followed by gain control and additive noise (**Fig 3**, panel 4). The result is a continuous time-varying signal in units of current (picoamps). While we use the same photocurrent model for all cones irrespective of size or location, the effect of the photocurrent depends on properties of the cones, due to the additive noise. Specifically, the signal-to-noise decreases more for larger cones than smaller cones, because large cones capture more photons and are subject to more downregulation before the additive noise.

**Midget RGC responses.** The fifth stage is spatial filtering by the mRGCs. We model the mRGCs in a rectangular array with each mRGC receptive field centered on a cone. We do not add further temporal filtering beyond that inherited from the photocurrent stage. We do not explicitly model spiking and its associated noise, but instead add independent Gaussian white noise to each RGC output at each time point. Unlike the photocurrent, where the noise is

implemented in ISETBIO according to a physiologically-informed model [80], the noise added in the mRGC layer is not constrained by a physiological model because the noise added by mRGCs (after accounting for noise inherited from prior stages) is less well known. For this reason, in additional simulations, we explore the effect of noise level in mRGCs, and find that while the mean performance declines with increasing noise (as expected), the differences between conditions are largely unaffected by noise level (**S4 Fig**). In the Discussion, we elaborate on the possible contribution of other aspects of retinal processing to polar angle asymmetries such as spatial subunits and spiking.

The mRGC layer has the same field-of-view as the cone array. Because we do not model rectification or spiking non-linearities, we do not separately model ON- and OFF-cells. Our mRGC receptive fields are 2D difference of Gaussian (DoG) models, approximating the shape of receptive fields measured with electrophysiology [81,82] (**Fig 3**, panel 5), based on parameters from macaque [83]. The width of the center Gaussian ($\sigma_c$, 1 sd) is ⅓ of the spacing between neighboring cones, and the surround Gaussian ($\sigma_s$) is 6x the width of the center. This creates an mRGC array with one mRGC per cone and where mRGC RFs overlap at 1.3 standard deviations from their centers, which matches the overlap of dendritic fields reported in human retina [55]. We compute the mRGC responses by convolving the cone absorptions with this mRGC DoG receptive field. Because the ratio of mRGCs to cones varies across the retina, we simulate differences in this ratio by subsampling the mRGC array (**Fig 4**). Thus, the mRGC density (cells/deg$^2$) is determined by both the cone array density and the cone-to-mRGC ratio, creating a 2D space of simulations.

**Behavioral inference.** The final stage of the computational observer model is the inference engine. For the main analysis, we use a linear support vector machine (SVM) classifier to discriminate stimulus orientation (clockwise or counter-clockwise from vertical) given the cone absorptions, cone photocurrent, or mRGC responses. We compute a weighted average across time for the output of each cell before running the classifier. This greatly reduces the dimensionality of the classifier input, and therefore speeds up computation time and reduces the number of trials needed for the classifier to learn optimal classification boundary. The weighting is proportional to the temporal filter in the photocurrent simulation, such that the time points with the highest weight in the filter has the largest contribution to the weighted average. Because we do not simulate eye movements or vary the phase of the stimulus, the only changes over time arise from the noise and temporal filtering by the photocurrent, and hence there is little to no loss of signal from averaging. The classifier trains and tests on the averaged responses for each stimulus contrast separately, where each contrast level results in a percent correct identified stimulus. The accuracy results are then fitted with a Weibull function to extract the contrast threshold at ~80%.

## The cone photocurrent and mRGCs have a large effect on orientation discrimination

We find large effects on performance of the computational observer when adding the cone photocurrent and the mRGC layers. For comparison, we ran the SVM decision maker either on the cone absorptions, the cone photocurrent, or the mRGC outputs while varying the cone density and the stimulus contrast. Consistent with our prior model [50], thresholds are low (~0.1–0.2%) when analyzed on the cone absorptions, and show only a small effect of cone density (**Fig 5A**). Thresholds increase sharply, about 5–10x, after the absorptions are converted to photocurrent (**Fig 5B**). This increase is due to noise in the photocurrent, consistent with prior results [51]. Surprisingly, the effect of cone density is also substantially increased, as seen in the greater spread of the psychometric functions. This is because the cones in the lower density

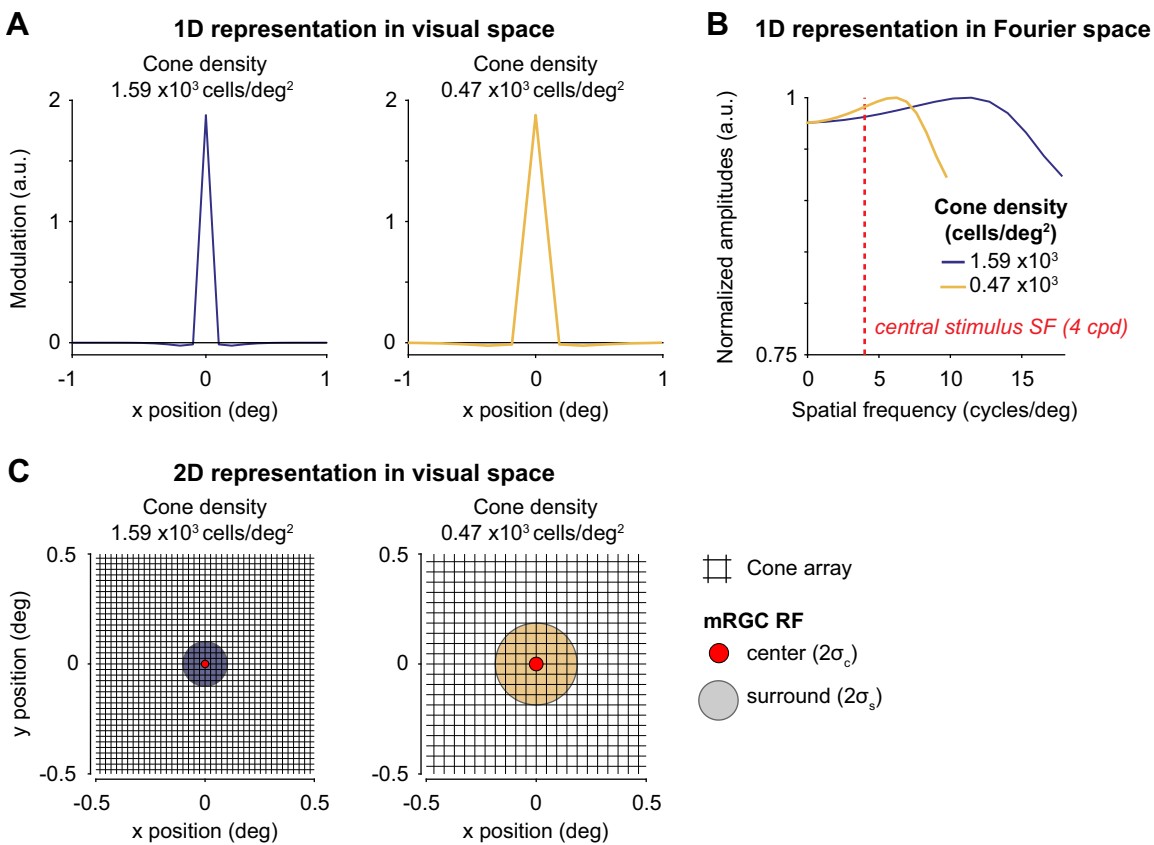

**Fig 4. Difference of Gaussians filters used to model mRGC layer.** Two mRGCs are illustrated for a 2x2˚ field-of-view mRGC array centered at 4.5˚ and 40˚ eccentricity. (A) 1D representation of two example mRGC layers in visual space. The mRGC responses are computed by convolving the cone image with the mRGC DoG RF, followed by adding noise, and subsampling the cone array to the corresponding mRGC density. Width for Gaussian center ($\sigma_c$) and surround ($\sigma_s$) are converted to units of degree. As the mRGC filters in our model are not rectified, they respond to both increments and decrements. Physiologically, this would require two cells (an ON and OFF cell), so we count each modeled mRGC location as two cells. Both panels show a mRGC:cone ratio of 2:1. (B) 1D representation of Difference of Gaussians in Fourier space. The Fourier representation illustrates the band-pass and unbalanced nature of the DoG (*i.e.*, non-zero amplitude at DC). Depending on the width/subsample rate, DoGs attenuate different spatial frequencies. However, at our peak stimulus frequency (4 cycles per degree, indicated with red dashed line) the two DoG filters vary a relatively small amount, preserving most stimulus information. Fourier amplitudes are normalized. Note that y-axis is truncated for illustration purposes. (C) 2D representation of two example mRGC layers shown in panel. Midget RGC DoG filters are zoomed into a 1x1˚ field-of-view cone array (black raster) centered at 4.5˚ (red center with purple surround) and 40˚ eccentricity (red center with yellow surround), corresponding to the 1D examples in panel A. Centers and surrounds are plotted at 2 standard deviations. For illustration purposes, only one mRGC is shown; the mRGC array in our computational observer model tiles the entire cone array.

retinal patches have larger apertures, resulting in greater photon capture, and hence more downregulation when converted to photocurrent. Over the 10-fold range of retinal densities, threshold vary by only about 1.4:1 for the absorptions, much less in contrast to about 5:1 for the photocurrent. The spatial filtering and late noise from the mRGCs further elevate thresholds, but at a fixed mRGC:cone ratio there is little change in the effect of cone density: the threshold vs density plot shows a vertical shift compared to the cone photocurrent, with about the same slope (**Fig 5C**).

We next quantified the effect of the mRGC:cone ratio on computational observer performance. We find that as the ratio increases, contrast thresholds decline (**Fig 6A**). The effect of the mRGC:cone ratio is largely independent of the cone density. For example, at any cone density, downsampling the mRGC density by 4x elevates thresholds by about 70% to 80%. The better model performance with more mRGCs comes from higher SNR, which arises because the

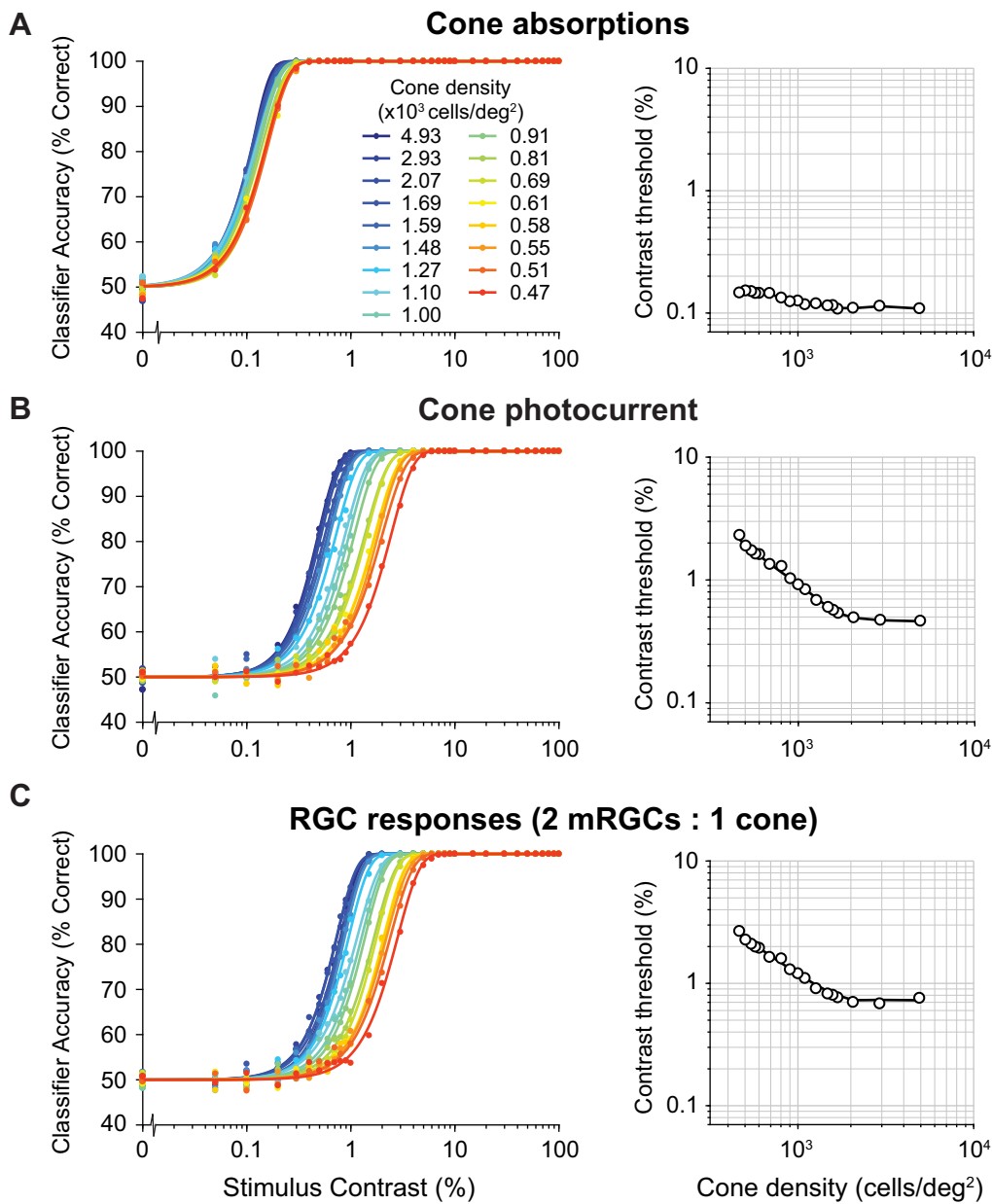

**Fig 5. Model performance for different computational stages.** Left column shows classifier accuracy as function of stimulus contrast. Data are from simulated experiments with 1,000 trials per stimulus class, using a model with a L-cone only mosaic varying in cone density. Data are fitted with a Weibull function. Contrast thresholds are plotted separately as a function of cone density in the right column. (A) Cone absorptions. Applying a linear SVM classifier to cone absorptions averaged across stimulus time points. (B) Cone photocurrent. Applying a linear SVM classifier to cone outer segment photocurrent responses, averaged across time weighted by a temporally delayed stimulus time course. This transformation of cone absorptions into photocurrent causes a ~10x increase in contrast thresholds, interacting with cone density (*i.e.*, Weibull functions are spaced out compared to cone absorptions). (C) RGC responses. Applying a linear SVM classifier to spatially filtered photocurrent with added white noise. This transformation causes an additional increase in contrast thresholds for all cone densities. Data show results for a fixed subsampling ratio of 2 mRGCs per cone.

signal is correlated across mRGCs (due to spatial pooling), whereas the noise added in the mRGC layer is independent. To visualize the space of predicted contrast thresholds as a function of cone density and mRGC:cone ratio, we plot model thresholds as a function of both

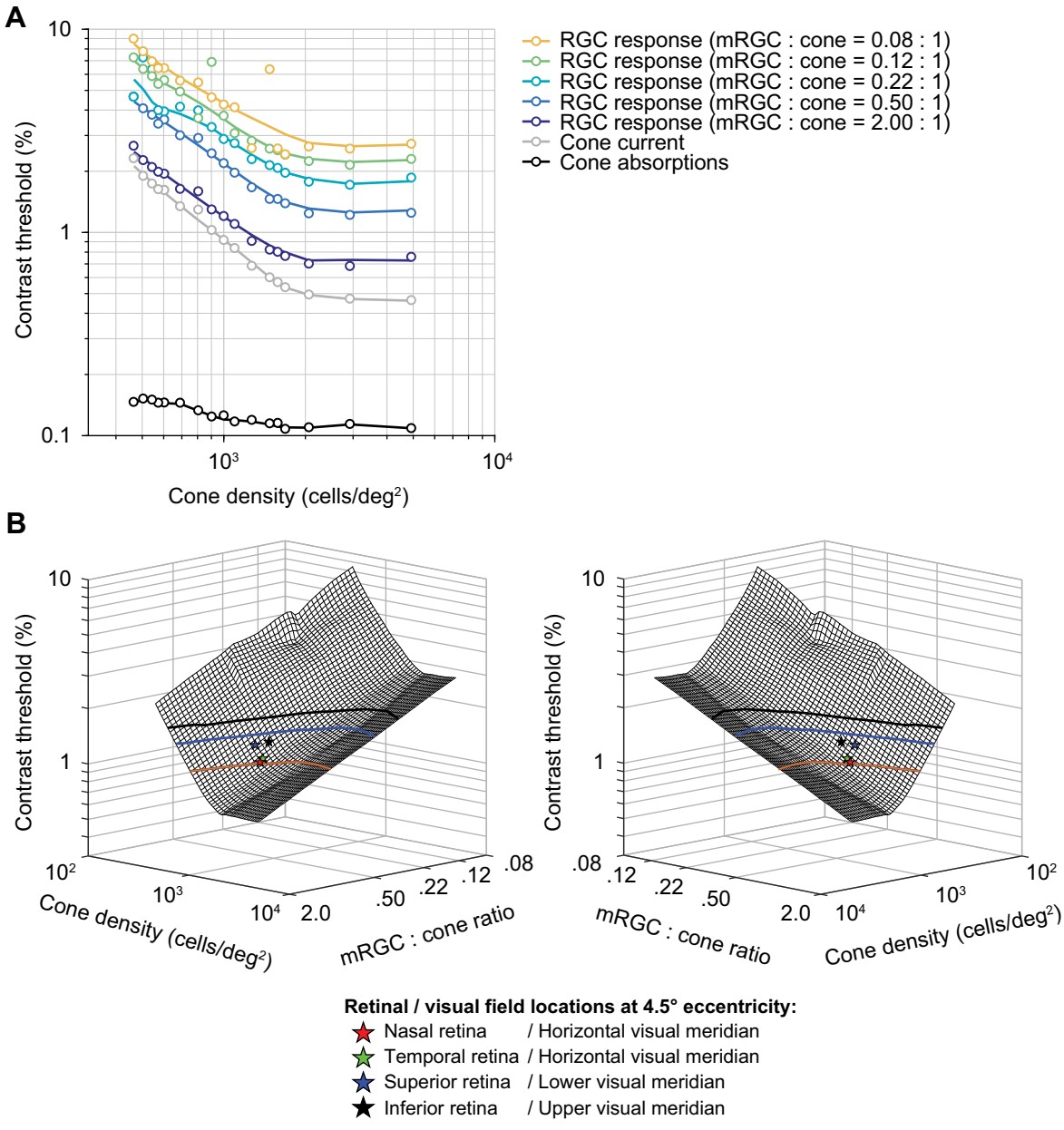

**Fig 6. The effect of spatial filtering properties by mRGCs on full model performance.** (A) Contrast thresholds as a function of cone density and mRGC:cone ratio. Data points are contrast thresholds for cone absorptions, cone photocurrent, and each mRGC:cone ratio separately (for psychometric functions see **S3 Fig**). Individual mRGC fits are slices of the 3D mesh fit shown in panel B. (B) Mirrored views of combined effect of cone density and mRGC:cone ratio on contrast sensitivity. The mesh is fitted with a locally weighted regression to 3D data: log cone density (x-axis) by log mRGC:cone ratio (y-axis) by log contrast thresholds (z-axis). Individual dots represent the predicted model performance for nasal retina or horizontal visual (red star), superior retina or lower visual (blue star), temporal retina or horizontal visual (green star) and inferior or upper visual (black star) meridian locations at 4.5° eccentricity (matched to stimulus eccentricity in [15]). Contour lines show possible cone densities and mRGC:cone ratios that would predict the same horizontal-vertical and upper/lower vertical-meridian asymmetry as observed in psychophysical data at 4.5˚ eccentricity. To do so, we scaled the difference in contrast threshold between the lower (blue) and upper (black) vertical visual meridian relative to the horizontal meridian to match the difference in behavior. Goodness of fit of 3D mesh fit is $R^2 = 0.96$.

independent variables (**Fig 6B**). This surface plot confirms the observation from the line plots (**Fig 6A**) that the effects of these two retinal factors—cone density and mRGC:cone ratio—have approximately independent, additive effects on model contrast threshold.

## Comparison between model and human contrast sensitivity

To compare model performance to human observers, we evaluate the model outputs for cone densities and mRGC:cone ratios that match the values on the different meridians according to the literature. We then compare these predicted thresholds to those obtained in a recent psychophysical experiment [15]. We also compare both the human data and the mRGC model data to two simplified models, one which omits the mRGCs and one which omits mRGCs and the conversion from isomerizations to photocurrent.

According to Curcio *et al.* [9], cone density at 4.5˚ eccentricity is ~1,575 cones/deg$^2$ on the horizontal retinal meridian (nasal: 1590 cones/deg$^2$, temporal: 1560 cones/deg$^2$), 1300 cones/deg$^2$ on the superior retinal meridian, and 1382 cones/deg$^2$ on the inferior retinal meridian. We combine these cone density values with the mRGC:cone ratios from the computational model by Watson [64], which ranges between 0.84 mRGCs per cone on the horizontal meridian (nasal: 0.87, temporal: 0.82), to 0.81 on the superior retina and 0.68 on the inferior retina.

Consistent with our previous report [50], we find that a model in which the pattern of photon absorptions is fed into the linear SVM classifier shows only a small effect of cone density (**Fig 7A**, left). Given the expected cone densities at the different polar angles at 4.5˚ eccentricity, the model predicts only about 5% higher sensitivity for the horizontal than vertical visual meridians, much less than the 40% difference found in behavioral experiments [15] (**Fig 7B**). The model also predicts almost no difference between upper and lower vertical visual meridian, whereas human sensitivity was found to be about 20% higher on the lower than upper vertical visual meridian. The overall sensitivity of the model observer (800–900) is considerably higher than human sensitivity (~30–50).

The conversion from cone absorptions to cone photocurrent reduces the sensitivity by about 4- to 5-fold, and increases the asymmetries. The linear SVM classifier performance

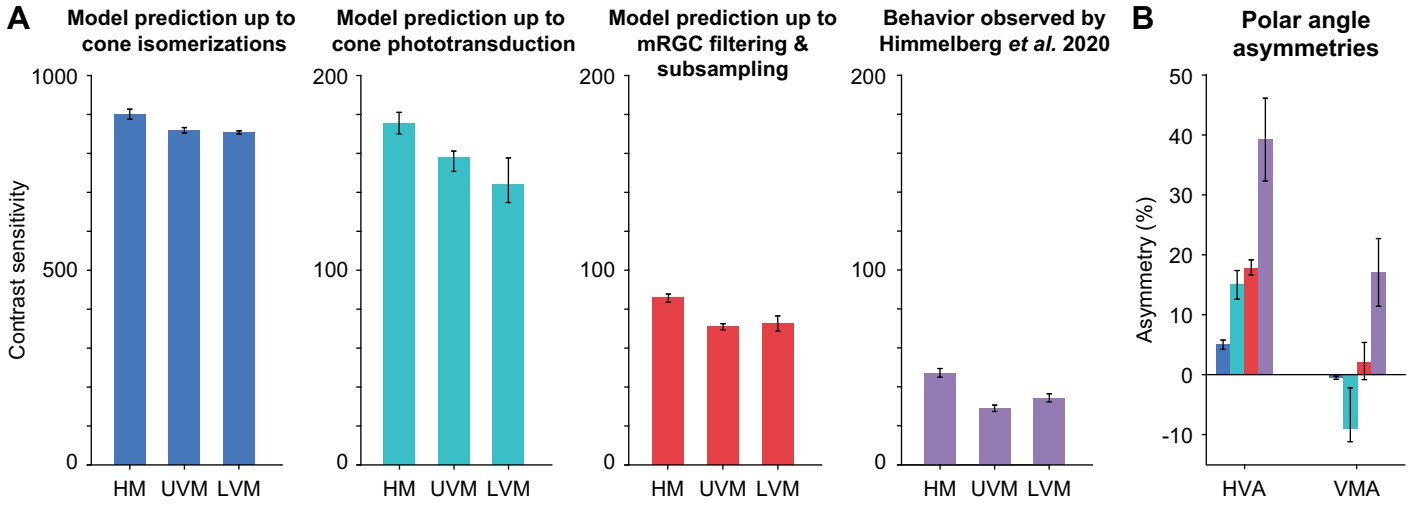

**Fig 7. Comparison of model performance to human performance.** (A) Contrast sensitivity predicted by computational observer model up to isomerizations in cones (blue), up to cone outer segment phototransduction (turquoise), up to spatial filtering and subsampling in mRGCs (red), and behavior observed (purple) by Himmelberg et al. (2020) using matching stimulus parameters. HM: horizontal meridian, UVM: upper visual meridian, LVM: lower visual meridian. Model prediction shows contrast sensitivity (reciprocal of contrast threshold) for stimuli at 4.5˚ eccentricity, with a spatial frequency of 4 cycles per degree. HM is the average of nasal and temporal meridians. Model error bars indicate simulation results allowing for uncertainty in the cone or mRGC density along each meridian (see Methods for details). Behavioral plots show group average results (*n* = 9) from Himmelberg et al. [15], and error bars represent standard error of the mean across observers. (B) Polar angle asymmetries for cone absorptions, photocurrent, mRGCs, and behavior. HVA: horizontal-vertical asymmetry. VMA: vertical-meridian asymmetry. Blue, turquoise, red, and purple bars match panel (A) and correspond to model prediction up to cone absorptions, cone photocurrent, mRGCs, human behavior. Error bars represent the HVA and VMA when using the upper/lower bound of predicted model error from panel A.

based on the cone photocurrent shows about 15% higher sensitivity for horizontal than vertical visual meridian, an asymmetry that is 3 times larger than that found in a model up to cone isomerizations. It also predicts about 9% higher sensitivity for upper vertical than lower vertical visual meridian (opposite to the pattern in human data). This is because the cone density is slightly higher for the upper than lower vertical visual meridian at this eccentricity (4.5˚).

Finally, the mRGC model brings overall performance closer to behavior, with sensitivity of about 70–90, and ~18% higher sensitivity for the horizontal than vertical visual meridian, predicting almost half the asymmetry found in behavior (~40%). The mRGC model also eliminates the advantage for upper over lower vertical visual meridian (now predicting slightly higher performance for the lower vs upper vertical), which is the same direction as the pattern observed in the human data.

Overall, our models show that although including an mRGC layer predicts polar angle asymmetries closer to behavior than a model up to cone absorptions or up to photocurrent, the biological variations in the spatial properties of mRGCs are not sufficient to fully explain differences in behavior. For example, the measured cone densities for the upper and lower vertical visual meridians are about 12% and 19% lower than for the horizontal. To predict the horizontal-vertical and vertical-meridian asymmetries as observed in human performance, and without further changing the mRGC:cone ratios, the cell densities would instead have to be ~37% and 30% lower than the horizontal. Alternatively, one could keep the cone densities fixed at the levels estimated by Curcio *et al.* [9], and instead vary the mRGC:cone ratio as observed by Watson [64]. In this case, the ratios would have to decrease from 0.81 to 0.52 for the lower vertical and 0.68 to 0.32 for the upper vertical visual meridian. If one decreased both the cone densities and the mRGC:cone ratios by tracing out the values along the nasal retinal meridian, one would need to increase eccentricity of a stimulus from 4.5˚ to 7.3˚ (upper vertical) or 6.3˚ (lower vertical) to match the behavioral asymmetries.

## Discussion

The visual system, from retina to subcortex to cortex, is organized in orderly maps of the visual field. But within each particular processing stage, the retinotopic map is distorted. Here we investigated the polar angle asymmetries in these spatial representations across three stages of the early visual pathway: cones, mRGCs, and V1 cortex. Our study revealed that both the eccentricity gradient (foveal bias) and polar angle asymmetries (HVA and VMA) in spatial representations are amplified from cones to mRGCs, and further amplified from mRGCs to early visual cortex. Additionally, we showed that although mRGC density has considerably polar angle asymmetries in the directions predicted by psychophysical studies, they are insufficient to explain observed differences in human's contrast sensitivity around the visual field.

### Linking behavior to eccentricity and polar angle asymmetries in visual field representations

For over a century, limits in retinal sampling were hypothesized to cause the fovea-to-periphery gradient in human visual performance [1,5,6]. Initial tests of this idea showed that the fall-off in cone density could explain some, but not all of the observed decrease in visual acuity [2,3,84–87]. Later, more detailed computational models, reported that mRGCs come closer in predicting the eccentricity-dependent decrease in achromatic contrast sensitivity and resolution, and conclude that mRGCs are sufficient to explain some aspects of behavior, such as spatial resolution and contrast sensitivity [88–94]. Similar to the retina, the cortical magnification factor in V1 has been linked to visual performance as a function of eccentricity, for example,

explaining differences in acuity [92,95,96], contrast sensitivity and resolution [20], visual search [97,98], and the strength of some visual illusions [99].

Conversely, polar angle asymmetries have rarely been considered. For instance, all above-mentioned studies either ignored the stimulus polar angle for analysis or limited measurements to a single meridian, usually the horizontal. Despite the fact that the existence of polar angle asymmetries in human early visual cortex was predicted based on behavior in the late 70's [19,20], further reports on polar angle differences have been scarce. One fMRI study reported a higher V1 BOLD amplitude for stimuli on the lower than the upper visual meridian [100] and two studies found more cortical surface area devoted to the horizontal than the vertical meridian [101,102]. Our recent studies suggest that V1 surface area is highly correlated to spatial frequency thresholds [78] and contrast sensitivity [103]. Yet several studies have assumed little to no polar angle differences in macaque V1 CMF [104,105] or did not account for polar angle differences in human V1 CMF [46,96] to explain differences in behavior. Computational models that include retinal and/or V1 sampling across visual space generally exclude polar angle asymmetries (*e.g.*, [106,107]). A few cases do incorporate polar angle asymmetries in the retinal ganglion cell distribution, but they assume that these asymmetries are not amplified in cortex [108–110].

## Early visual cortex does not sample the retina uniformly

It is well documented that the convergence of cones to retinal ganglion cells varies with eccentricity (*e.g.*, see [91]). In the fovea of both primates and humans, there is one cone per pair of bipolar cells and pair of midget RGCs, with pairs comprised of an "ON" and an "OFF" cell. In contrast, in the periphery, there are many cones per pair of bipolar cells and midget RGCs, with the ratio depending on the eccentricity. In the far periphery, there can be dozens of cones per ganglion cell [9].

It has been long debated whether V1 further distorts the visual field representation, or if V1 samples uniformly from RGCs, as reviewed previously [71,72]. Our analysis showed more cortical surface area devoted to the fovea than the parafovea and to the horizontal than vertical meridian, supporting previous findings using retinotopy informed by anatomy [101] and functional MRI [78,102,103,111]. Importantly, these eccentricity and polar angle non-uniformities are larger in V1 than they are in mRGC density, in agreement with findings from monkey [61,73–75,112,113]. Whether these non-uniformities arise in cortex, or depend on the mapping from retina to LGN, LGN to V1, or both, is a question of interest in both human [114,115] and monkey [116–120], but beyond the scope of this paper. The implication of the increased spatial non-uniformities in the cortical representation is that cortex cannot be understood as a canonical wiring circuit from the retina repeated across locations.

Because visual field distortions are larger as a function of eccentricity than polar angle, one might surmise that polar angle asymmetries contribute little to visual performance. Even though the polar angle asymmetries are smaller than the eccentricity effects, they can in fact be large. For example, within the central eight degrees, the surface area in V1 is about 60% larger for the horizontal meridian than the vertical meridian [78]. Given that virtually all visual tasks must pass through V1 neurons, these cortical asymmetries are likely to have a large effect on perception. The number of cortical cells could be important for extracting information quickly [121], for increasing the signal-to-noise ratio, and for tiling visual space and visual features (*e.g.*, orientation, spatial frequency) more finely [122]. To know how the number of V1 neurons affect performance, there is a need for a computational model that explicitly links cortical resources to performance around the visual field.

## Temporal summation in cone photocurrent accentuates polar angle asymmetries

We found one physiological factor in the retina—gain control in the cone photocurrent—that appears to accentuate the polar angle asymmetries. This is because at matched eccentricities, cone density varies with polar angle (*i.e.*, cone density is higher on the horizontal meridian), and cone aperture size varies inversely with density. Specifically, at lower densities, the apertures are larger, capturing more photons per receptor. As a result of the higher absorption rates, there is greater downregulation of the photocurrent gain. Cottaris *et al.* [51] observed in their modeling work that the lower gain in the photocurrent for larger cones caused a reduction in the signal-to-noise ratio. In their simulations, this resulted in sensitivity loss for stimuli that extended further into the periphery. In our simulations, lower density results in lower sensitivity, therefore contributing to the difference in performance as a function of polar angle.

Overall, while adding a photocurrent stage decreases overall thresholds, bringing them closer to human performance (especially for simulations with low cone density mosaics), it still leaves a large gap between the predicted and observed psychophysical asymmetries as a function of polar angle. Moreover, the photocurrent model does not explain any of the vertical meridian asymmetry, as cone density, and presumably aperture size, do not differ between lower and upper vertical meridian in a way that matches behavior.

## Model limitations

Despite implementing known facts about the eye, our model, like any model, is a simplification. The lack of comprehensiveness trades off with interpretability. For this model, we make the trade-off between complexity and understanding by treating a local patch of mRGCs as a linear, shift-invariant system (*i.e.*, a spatial filter). As several components of the model here are identical to our previous model, we will focus on the limitations of those components that are different (addition of cone photocurrent and mRGC layer, and exclusion of eye movements), and refer to Kupers, Carrasco, and Winawer [50] for model limitations related to the pathways from scene to cone absorptions and the inference engine.

**Spatial properties: Uniform sampling within a patch and subunits.** Hexagonal cone arrays that include within-patch density gradients have been implemented in ISETBIO by Cottaris *et al.* (e.g., [51,67]). Nonetheless, our mRGC layer is implemented as a rectangular patch of retina, initially with the same size as the cone mosaic. This allows for filtering by convolution and then linear subsampling to account for mRGC density, making the model computationally efficient. We do not incorporate several known complexities of RGC sampling in the retina: (i) density gradients within a patch, (ii) irregular sampling, and (iii) spatial RGC subunits. (i) Given our relatively small patch size (2x2˚ field-of-view) in the parafovea (centered at 4.5˚), the change in density across the patch would be small (~10%). We found that a much larger change in mRGC density (spanning a 5-fold range) had only a modest effect on performance of our observer model, so it is unlikely that accounting for a small gradient within a patch would have significantly influenced our results. (ii) Given the relatively low spatial frequency content of our test stimulus (4 cycles per degree), it is unlikely that irregular sampling would have resulted in a substantial difference from the regular sampling we implemented. (iii) Our low spatial frequency test stimuli also reduce concerns of omitting spatial subunits [123–126], as these non-linearities are most likely to be important for stimuli at high spatial frequencies (reviewed by [127]). Moreover, we showed for our linear RGC filters that sensitivity differences are only large at high spatial frequencies (around 8 cycles per degree and higher); even when receptive field sizes differ by a factor of 3 (as shown in **Fig 4B**). Hence for the relatively low spatial frequency stimuli modeled here, the detailed spatial properties that

we excluded would likely not have large enough effects to make up the difference between the predicted model performance and human behavior.

**Temporal properties and eye movements.** In contrast to our previous work [50], our current model includes temporal integration but omits fixational eye movements and multiple cone types. The omission of eye movements made the model more tractable and the computations more efficient. We think this omission is unlikely to have a large effect on our results. In recent related work, it was shown that fixational eye movements combined with temporal integration resulted in spatial blur and degraded performance, causing a loss in contrast sensitivity up to a factor of 2.5 [51]. However, the largest losses were for stimulus spatial frequencies over 8 cycles per degree, with little loss from eye movements for stimuli with lower peak spatial frequency (2–4 cycles per degree). Given that the spatial frequency of our test stimulus falls within this range, the influence of fixational eye movements on the computational observer performance would have been modest.

**Noise implementation.** Our expectation was that the largest effect of mRGCs on performance as a function of polar angle would arise from variation in cell density: where mRGC density is higher, SNR will be higher, thus performance will be better. This effect of density on performance emerged in our simulations from the noise added after spatial filtering, before subsampling: without this additional noise component, the spatial filtering of the mRGC would just be a linear transform of the cone outputs, which would have little or no effect on performance of a linear classifier. We simulated this late noise as additive Gaussian noise rather than the stochastic nature of spiking, as we were not trying to fit spiking data but rather predict behavior. While we also did not build in correlated noise between RGCs (*e.g.*, [128]), there is nonetheless some shared noise in our mRGCs due to common inputs from cones, which is the major source of noise correlations in RGCs [129]. Moreover, we found that the general pattern of model performance was unchanged over a large range of noise levels (up to an overall scale factor in performance), suggesting that the effect of density is likely to hold in many noise regimes.

**Other retinal cell types.** Midgets are not the only retinal ganglion cells that process the visual field. Parasol (pRGCs) and bistratified retinal ganglion cells are less numerous but also cover the entire retina. pRGCs are the next most common retinal ganglion cells, and have generally larger cell bodies and dendritic field sizes than mRGCs, both increasing with eccentricity [54]. These differences cause parasols to be more sensitive to relative contrast changes and have higher temporal resolution, with the consequence of losing spatial resolution [130]. For this reason, the small mRGCs are much more likely to put a limit on spatial vision, and thus our model does not include pRGCs.

The discussion above raises the question, had we incorporated more known features of the retina in our model, would the model make predictions more closely matched to human performance? We think it is unlikely that doing so would fully explain the observed asymmetries in behavior, because we measured substantially larger asymmetries in cortex than in retina. If the retinal simulations entirely accounted for behavior, this would leave no room for the additional cortical asymmetries on behavior.

## A case for cortical contributions to visual performance asymmetries

Recent retinal modeling of contrast sensitivity in the fovea showed that very little information used for behavior seems to be lost from the retinal output [51]. This may not be the case for the parafovea and periphery. Incorporating temporal properties of phototransduction and spatial properties of mRGC followed by additive noise could explain about half the differences in behavior of HVA and ~1/6 of VMA. These differences indicate a contribution from

downstream processing, such as early visual cortex. V1 cortex has several characteristics that suggest a tight link between cortical topography and polar angle visual performance asymmetries. Hence a model that incorporates properties of early visual cortex is likely to provide a substantially better account of polar angle asymmetries in behavior than one that only incorporates properties of the eye. We have not developed such a model but outline some of the reasons that cortex-specific properties are important for explaining polar angle asymmetries.

First, the representation of the visual field is split across hemispheres in visual cortex along the vertical, but not horizontal meridian. This split may require longer temporal integration windows for visual input that spans the vertical meridian, as information needs to travel between hemispheres. For example, the response in the left visual word form area is delayed by ~100 ms compared to the right visual word form area when presenting a stimulus in the left visual field [131]. Longer integration windows may in turn impair performance on some tasks, as eye movements during integration will blur the representation. Longer integration time of visual information spanning the vertical meridian is consistent with behavior, as accrual time is slower when stimuli are presented at the vertical than the horizontal meridian [38]. Interestingly, the hemispheric split is not precise: there is some ipsilateral representation of the visual field along the vertical meridian in early visual cortex. The amount of ipsilateral coverage is larger along the lower than upper vertical meridian and increases from 1–6˚ eccentricity [132]. It is possible that the split representation affects performance for stimuli on the vertical meridian (contributing to the HVA), and that the asymmetry in ipsilateral coverage between the lower and upper vertical meridian contributes to the VMA.

Second, there is good correspondence between the angular patterns of asymmetries in V1 cortex and behavior. Polar angle asymmetries in the CMF of early visual cortex are largest along the cardinal meridians (*i.e.*, horizontal vs vertical and upper vertical vs lower vertical). The asymmetries gradually fall-off with angular distance from the meridians [78]. This gradual decrease in polar angle asymmetry in cortex parallels the gradual decrease in contrast sensitivity [12,29,30] and spatial frequency sensitivity [16] with angular distance from the cardinal meridians. Measurements of cone density and retinal ganglion cell density have emphasized the meridians, so there is less information regarding how the asymmetries vary with angular distance from the meridians.

Third, there is good correspondence between cortical properties and behavior in the domain of spatial frequency and contrast sensitivity. Polar angle asymmetries in spatial frequency sensitivity observed by Barbot *et al.* [16] parallel spatial frequency tuning in V1 cortex. Specifically, fMRI measurements show that in V1, in behavior spatial frequency thresholds are higher on the horizontal than vertical visual meridian [16] and the preferred spatial frequency tuning is higher along the horizontal meridian than vertical visual meridian [133]. Additionally, polar angle asymmetries in contrast sensitivity covary with polar angle asymmetries in V1 cortical magnification [103]: Observers with larger horizontal-vertical asymmetries in contrast sensitivity (*i.e.*, better performance on the horizontal vs vertical visual meridian at matched eccentricities), tend to have larger horizontal-vertical asymmetries in V1 cortical magnification at corresponding locations in the visual field.

Fourth, polar angle asymmetries in behavior are maintained when tested monocularly [12,16], but thresholds are slightly higher compared to binocular testing (at least for spatial frequency sensitivity [16]). Higher thresholds (*i.e.*, poorer performance) show that performance benefits from combining information of the two eyes, as twice the amount of information increases the signal-to-noise ratio [134]. This summation is likely to arise in early visual cortex, as V1 is the first stage in the visual processing pathways where information of the left and right visual field merges [135–137].

## Conclusion

Overall, we have shown that the well-documented polar angle asymmetries in visual performance are associated with differences in the structural organization of cells throughout the early visual pathway. Polar angle asymmetries in cone density are amplified in downstream processing, from cones to RGCs and again from RGCs to early visual cortex. Further, we have extended our computational observer model to include temporal filtering when converting cone absorptions to photocurrent and spatial filtering of mRGCs, and found that both contributions, although larger than those of cones, are far from explaining behavior. In future research, we will aim to integrate cortical data within the computational observer model to explain whether a significant amount of the polar angle asymmetries can be accounted for by the organization of cortical space in early visual cortex.

## Methods

### Reproducible computation and code sharing

All analyses were conducted in MATLAB (MathWorks, MA, USA). Data and code for our previously published and extended computational observer model, including density computations and figure scripts, are made publicly available via the Open Science Framework at the URL: https://osf.io/mygvu/ (previously published) and https://osf.io/ywu5v/ (this study).

### Data sources

Data on cone density, midget RGC density, and V1 cortical surface area previously published or from publicly available analysis toolboxes. Both cone and mRGC densities were computed as cells/deg$^2$ for 0–40˚ eccentricities (step size 0.05˚), at the cardinal meridians (0˚, 90˚, 180˚, and 270˚ polar angle, corresponding to nasal, superior, temporal, and inferior retina of the left eye. **Fig 1** contains averaged cone and mRGC densities across all meridians as a function of eccentricity. **Fig 2** contains cone and mRGC densities converted to visual field coordinates, where the horizontal visual field meridian is the average of nasal and temporal retina, upper visual field meridian corresponds to the inferior retina and lower visual field meridian to the superior retina.

**Cone density.** Cone density data for the main results were extracted from post-mortem retinal tissue of 8 human retina's published by Curcio *et al.* [9] using the analysis toolbox ISETBIO [65–67], publicly available via GitHub (https://github.com/isetbio/isetbio).

Cone density in **S1 Fig** shows two datasets computed by two analysis toolboxes. To extract post-mortem data from Curcio *et al.* [9], we either use ISETBIO or the rgcDisplacementMap toolbox [76], publicly available at GitHub (https://github.com/gkaguirrelab/rgcDisplacementMap). A second cone density dataset comes from an adaptive optics study published by Song *et al.* [10]. From this work, we use "Group 1" (young individuals, 22–35 years old) implemented in ISETBIO.

**Midget retinal ganglion cell receptive field density.** Midget RGC density for the main results were computed with the quantitative model by Watson [64] implemented in ISETBIO. This model combines cone density data from Curcio *et al.* [9], mRGC cell body data from Curcio and Allen [53] and the displacement model by Drasdo *et al.* [57], to predict the midget RGC receptive fields (RFs).

Midget RGC data in **S1 Fig** computes mRGC density with two computational models: Watson [64] from ISETBIO and the displacement model by Barnett and Aguirre [76] implemented in the rgcDisplacementMap toolbox.

**Cortical magnification factor in early visual cortex.** To quantify the fovea-to-periphery gradient in the V1 cortical magnification factor (CMF), we used the areal CMF function published in Horton and Hoyt [68] for 0–40˚ eccentricity (**Fig 1**). Because this function does not make separate predictions for the cardinal meridians (**Fig 2**), we used data from the Human Connectome Project (HCP) 7 Tesla retinotopy dataset (*n* = 163), which were first published by Ugurbil, van Essen, and colleagues [138,139] and analyzed with population receptive field models by Benson *et al.* [79]). V1 CMF surface area data are from Benson *et al.* [78] segmented into bins using hand-drawn ROIs from Benson *et al.* [140] and computed as follows.

To compute V1 CMF from retinotopy data, we used the extracted surface area for ±10˚ and ±20˚ wedge ROIs centered on the cardinal meridians in each individual's hemisphere. The wedges on the horizontal, dorsal, and ventral locations represented the horizontal, lower, and upper visual field meridians respectively. Wedge ROIs were computed in the following steps: First, area V1 and V2 were manually labeled with iso-eccentricity and iso-polar angle contour lines using the measured retinotopic maps of each hemisphere [140]. Second, for each cardinal meridian and each 1˚-eccentricity bin, we calculated the mean distance along the cortex to reach a 10˚ or 20˚ polar angle. All vertices that fell within the eccentricity bin and polar angle distance were included in the particular ROI. We computed wedge strips, rather than an entire wedge or line, to avoid localization errors in defining the exact boundaries.

The wedges were separated into 5 eccentricity bins between 1–6˚ (1˚ step size) using the hand-drawn ROIs from Benson *et al.* [140], marking eccentricity lines at 1˚, 2˚, 4˚, and 7˚. The 3˚, 5˚ and 6˚ eccentricity lines were deduced from the 2˚, 4˚ and 7˚ lines using isotropic interpolation (independently for ±10˚ and ±20˚ wedge ROIs, for more details see Benson *et al.* [78]), and hence are likely to be less accurate than the data points at the exact hand-drawn eccentricity lines. The cortical surface area (mm$^2$) was summed across hemispheres within each subject and divided by the visual field area (deg$^2$). For each eccentricity bin and cardinal meridian, mean and standard error V1 CMF were computed from bootstrapped data across subjects (1,000 iterations). Mean data for each cardinal meridian were fit with a linear function in log-log space (*i.e.*, power law function in linear coordinates) for 1–6˚ eccentricity.

The initial ROIs used for the upper and lower vertical meridian included both V1 and V2 sections of the vertical meridian, and therefore contain twice as much visual area as the horizontal ROI. To have a fair comparison between the horizontal and upper and lower visual field ROIs, we corrected the upper and lower ROIs as follows. For each subject and eccentricity bin, we computed a vertical surface area ROI (with both upper and lower visual fields) that excluded V2 sections of the vertical meridian. When summed over both hemispheres, this vertical ROI has a size comparable to the horizontal ROI. We then calculated a scale factor for each subject and eccentricity, by dividing the vertical ROI by the sum of upper and lower surface area ROIs. This scale factor was on average ~0.5. To get the corrected V1 CMF, we multiplied the scale factor to corresponding ventral and dorsal surface areas and divided by the corresponding visual field area. By scaling dorsal and ventral ROIs to only include the V1-side, we made the assumption that V2 is approximately the same size as V1. These vertical ROIs may be slightly less precise than the horizontal meridian ROI and affect the horizontal-vertical asymmetry (HVA). We did not compare differences in pRF sizes for the cardinal meridians.

Although the narrower ±10˚ wedge ROIs are in closer correspondence to the single line estimations of cone and mRGC density, we use ±20˚ wedge ROIs in **Fig 2** as those data are more robust. This is because narrow wedge ROIs are prone to overestimation of the vertical meridian surface, caused by ipsilateral representations near the boundaries. Such ipsilateral representations are sometimes incorrectly counted as part of the ±20˚ ROI for the ipsilateral hemisphere, instead of as part of the ±10˚ ROI for the contralateral hemisphere, and this effect

is exacerbated for smaller wedges. We visualize V1 asymmetries for both ±10˚ and ±20˚ wedge ROI **S1 Fig**.

### Convergence ratios

The cone:mRGC ratio was computed by dividing mRGC density (cells/deg$^2$) by cone density (cells/deg$^2$) for 0–40˚ eccentricity, in 0.05˚ bins. The mRGC:CMF ratio was computed in cells/mm$^2$. When comparing mRGC density to Horton and Hoyt's CMF prediction, mRGC density (cells/deg$^2$) was divided by V1 CMF (deg$^2$/mm$^2$) for 0–40˚ eccentricity, in 0.05˚ bins. When comparing HCP's retinotopy CMF to mRGC density, mRGC density was restricted to 1–6˚ eccentricity, and divided by the power law functions fitted to the V1 CMF. To compute the transformation ratios relative to horizontal visual field meridian for cone:mRGC or mRGC:V1 CMF ratios in **S2 Fig**, we divide the lower and upper visual field transformation ratio separately by the horizontal visual field transformation ratio.

### Asymmetry computation

Polar angle asymmetries between meridians for cone density and mRGC density were calculated as percent change in retinal coordinates as in Eqs 1 and 2, and then converted to visual field coordinates (*i.e.*, nasal and temporal retina are left and right visual field meridians, and superior and inferior retina are lower and upper visual field meridians):

$$Horizontal\ Vertical\ Asymmetry = 100 \cdot \frac{mean(nasal, temporal) - mean(superior, inferior)}{mean(nasal, temporal, superior, inferior)} \quad (1)$$

$$Vertical\ Meridian\ Asymmetry = 100 \cdot \frac{superior - inferior}{mean(superior, inferior)} \quad (2)$$

Polar angle asymmetries in V1 CMF and behavior were computed with the same equations, but for visual field coordinates.

### Computational observer model

The computational observer uses and extends a published model [50]. The extensions include (1) a phototransduction stage in the cone outer segment (transforming absorptions to photocurrent) and (2) a midget RGC layer (transforming photocurrent to mRGC responses) between the cone isomerization stage and the behavioral inference stage. To compensate for the increase in computational load and to keep the model tractable, we also made two simplifications: We used an L-cone only mosaic (instead of L-, M-, S-cone mosaic), and removed any stimulus location uncertainty by omitting fixational eye movements and stimulus phase shifts within a single stimulus orientation. With our extended model, we generated new cone absorption and photocurrent data using a fixed random number generator.

Given that several stages of the model are identical to those to the previous study, we refer to those methods on *Scene radiance*, *Retinal irradiance*, and *Cone mosaic and absorptions*. Unlike in our previous study [50], we did not vary the level of defocus in the *Retinal irradiance* stage nor the ratio of different cone types within a cone mosaic.

**Stimulus parameters.** The computational model simulates a 2-AFC orientation discrimination task while varying stimulus contrast. The stimulus parameters are chosen to match the baseline condition of the psychophysical study by Himmelberg *et al.* [15], whose results have replicated the psychophysical study used for comparison in our previous computational observer model [13]. The recent psychophysics experiment used achromatic oriented Gabor

patches, ±15˚ oriented from vertical, with a spatial frequency of 4 cycles per degree. Stimuli were presented at 4.5˚ iso-eccentric locations on the cardinal meridians, with a size of 3x3˚ visual angle (σ = 0.43˚) and duration of 120 ms. These stimulus parameters were identical to those the model, except for size, duration, and phase randomization of the Gabor. The simulated stimulus by the model was smaller (2x2˚ visual angle (σ = 0.25˚), shorter (54-ms on, 2-ms sampling) followed by a 164-ms blank period (mean luminance). We simulated these additional time points without a stimulus because photocurrent data are temporally delayed (see next section on *Photocurrent*). There was no stimulus onset period, and the phase of the Gabor patches were identical across all trials (90˚). Instead of simulating 5 experiments with 200 trials per stimulus orientation as in our previous paper, we simulated one experiment with 5x more trials (*i.e.*, 1,000 trials per stimulus orientation, 2,000 trials in total) to ensure that our behavioral inference stage had sufficient number of trials to successfully learn and classify stimulus orientation. To assure psychometric functions with lower and upper asymptotes, stimulus contrasts ranged from 0–100%.

**Photocurrent.**    After the cone isomerization stage, we applied ISETBIO's built-in *osLinear* photocurrent functionality implemented by Cottaris *et al.* [51] to our cone absorption data (separate for each simulation varying in cone density). This photocurrent stage converts cone excitations into photocurrent in pA in a linear manner (in contrast to the *osBiophys* functionality in ISETBIO which contains a more complex and computationally intensive biophysical model to calculate cone current).

The phototransduction stage takes the cone absorptions and applies three computations. First, it convolves cone absorptions trials with a linear temporal impulse response specific to L-cones (see **Fig 3**, panel in between absorptions and photocurrent stage). This temporal filter delays and blurs the cone photocurrent in time. Second, photocurrent gain is downregulated by light input, for instance due to increased luminance levels or larger cone apertures. Third, photocurrents are subject to an additional source of white Gaussian noise, which are determined by photocurrent measurement by [80] (for more details, see Cottaris *et al.* [51]). This resulted in a 4D array with *m* rows by *n* columns by 109 2-ms time points by 2,000 trials.

Because our simulated experiments do not contain any uncertainty about the stimulus location (no fixational eye movements or stimulus phase randomization), we were able to average both cone absorptions and photocurrent data across stimulus time points. We computed mean cone absorption data by taking the average across the first 54 ms (ignoring the time points without stimulus). For mean cone photocurrent data, we took a weighted mean across all 218 ms time points using a temporally delayed stimulus time course. This time course was constructed by convolving the stimulus on-off boxcar with the temporal photocurrent filter. This resulted in a 3D array with time-averaged cone photocurrent *m* rows by *n* columns by 2,000 trials.

**Midget RGC layer.**    Prior to the mRGC layer, Gabor stimuli were simulated as spectral scene radiance from a visual display, passed through the simulated human optics, subject to isomerization and phototransduction by the cones in a rectangular mosaic (2x2˚ field-of-view) and saved as separate files for each stimulus contrast. The mRGC layer loaded the simulated 2D cone absorptions and photocurrent data.

The mRGC layer was built as a rectangular array, with the identical size mosaic as the cone mosaic (2x2˚). Spatial summation by RGC RFs was implemented as 2D Difference of Gaussians (DoG) filters [81,82]. The DoG RF was defined on a support of 31 rows by 31 columns. The DoG size was based on Croner and Kaplan [83]: the standard deviation of the center Gaussian ($\sigma_c$) was 1/3 times the cone spacing and the standard deviation of the surround Gaussian ($\sigma_s$) was 6 times the center standard deviation. The center/surround weights were 0.64:0.36, hence unbalanced. These parameters create neighboring DoG RFs that overlap at 1.3

standard deviation from their centers, approximating RGC tiling in human retina based on overlap of dendrites fields [55]. The support of the DoG filter did not change size, however, because the mRGC array is matched to the cone array and cone density affects cone spacing (*i.e.*, a lower cone density results in a sparser array), the width of the DoG varies with cone density and can be expressed in units of degree visual angle (*i.e.*, scaling with the number of cones per degree within the cone array).

In the primate fovea, there is one ON and one OFF mRGC cell per cone, for a ratio of 2 mRGCs per cone. Unlike in the eye, our model mRGCs are not rectified, hence one of our mRGCs can signal either increments or decrements. For comparison to the literature, we multiply our mRGC counts by 2. We do not model ON- and OFF-center mRGCs separately, but rather consider one linear mRGC (no rectification) as a pair of rectified ON- and OFF-centers. For example, we consider an mRGC layer with no subsampling as having an mRGC:cone ratio of 2:1 (2 mRGCs per cone). The mRGC:cone ratios, counted in this way, were 2:1, 0.5:1, 0.22:1, 0.125:1, 0.08:1. The highest ratio (2:1) is similar to the observed in the fovea and the lowest ratio (0.08:1) is similar to the observed at ~40˚ eccentricity [64]. We tested a wide range of ratios because the purpose of the modeling was to assess how variation in mRGC density affects performance. The relationships between cone density and performance, or between mRGC:cone ratio and performance, are more robustly assessed by testing a wide range of parameters.

The spatial computations of the mRGC layer were implemented in three stages. In the first stage, the 2D DoG filter was convolved with each time-averaged 2D cone photocurrent frame separately for each trial. The photocurrent images were padded to avoid border artifacts. We padded the array with the mean of the photocurrent cone array, where the padding doubled the width and height of the array. The post-convolution array maintained the same size as the cone array without padding.

In the second stage, white Gaussian noise was added to each time point of the filtered cone photocurrent response, sampled from a distribution with a standard deviation of 1. This noise level was determined after testing a range of values showed that doubling or halving the width of the Gaussian only scaled the absolute performance levels, not the effect as a function of cone density or mRGC:cone ratios (for results using a standard deviation of 0.5 and 2, see **S4 Fig**). We added noise to our mRGC responses at this stage, because our mRGC layer without noise would perform a linear transform of the photocurrent responses (linear filtering and linear subsampling). A transform that a linear support vector machine classifier should be able to learn the optimal hyperplane with enough training trials to "untangle" the two stimulus classes. This would mean that our model would not predict any loss of information introduced by the mRGC layer, the effect we are most interested in. Had we used a limited number of trials instead, our model would have performed suboptimal and showed differences in classification accuracy. In such case, it would be difficult to distinguish the extent to which these performance differences are caused by spatial variations in mRGCs on visual performance versus the general ability of the SVM algorithm.

In the third stage, the filtered cone responses were linearly subsampled. This was implemented by resampling each row and column of the filtered cone responses with a sample rate equal to the mRGC:cone ratio. For instance, an array with an mRGC:cone ratio of 0.5:1 samples from every other cone. The mRGCs are centered on the cones, limiting the resampling of filtered cone responses to integer numbers of cones. These spatially filtered and subsampled responses are the mRGC responses in arbitrary units, as we added an arbitrary level of Gaussian white noise on the filtered photocurrent responses and did not implement spiking non-linearity in this transformation.

**Simulated experiments.** A single simulated experiment had a total of 64,000 trials: 2,000 trials per contrast level, 1,000 clockwise and 1,000 counter-clockwise. Stimulus contrast was systematically varied from 0 to 100% Michelson contrast, using 32 contrast levels. The cone mosaic was identical across contrast levels, only including L-cones, cone density and cone spacing. There were no eye movements. Cone absorptions and photocurrent simulations used a fixed random number generator seed. Data from a single contrast level were represented as a 4D array ($m$ rows by $n$ columns by 218 time points by 2,000 trials). The size of the $m$ by $n$ frame depended on the defined subsampling ratio used for the mRGC layer.

This single experiment was repeated for 17 different cone mosaics, which varied systematically in cone density and spacing. The cone density variation was implemented by simulating cone mosaics at different eccentricities, ranging from a density as high as at the 1˚ (4.9 x$10^3$ cells/deg$^2$) to as low as at 40˚ eccentricity on the horizontal meridian (0.047 x$10^4$ cells/deg$^2$). This resulted in a total of 1,088,000 simulated trials (64,000 trials x 17 cone densities).

Simulated experiments for each of the 17 different cone densities were averaged across time, resulting in a 3D array ($m$ rows by $n$ columns by 2,000 trials). In the mRGC layer, each 3D array was spatially subsampled by 5 different mRGC:cone ratios. This resulted in a total of 5,440,000 simulated trials (64,000 trials x 17 cone densities x 5 ratios).

**Inference engines.** The simulated trials were fed into an inference engine. The task of the inference engine was to classify if a trial contained a clockwise or counter-clockwise oriented Gabor stimulus given the cone or mRGC responses. Classification was performed separately for every 2,000 trials, *i.e.*, separately for each contrast level, cone density, and mRGC:cone ratio.

We used a linear SVM classifier as implemented in MATLAB's *fitcsvm* with 10-fold cross-validation and built-in z-scoring. This procedure is identical to our previously published model [50]. In contrast to our previous model implementation, we did not transform each 2D frame of mRGC responses to the Fourier domain and did not discard phase information prior to classification, because the stimulus was static and did not contain any uncertainty about stimulus location nor simulated fixational eye movements. The mRGC responses were concatenated across space, resulting in a matrix of 2,000 trials by mRGC responses. The order of the trials within this vector was randomized and fed into the linear SVM classifier with a set of stimulus labels. The classifier trained its weights on 90% of the trials, and tested on the 10% left-out trials. This resulted in accuracy (percent correct) for each given contrast level, cone density and ratio.

Accuracy data for a single simulated experiment were fitted with a Weibull function to extract the contrast threshold. The threshold was defined as the power of 1 over the slope of the Weibull function, which comes out approximately ~80% correct, given that chance is 50% for a 2-AFC task and our slope was defined as $\beta$ = 3.

**Comparing model performance to behavior.** To quantify the contribution of the spatial filtering by mRGCs, we compared the model performance to behavior reported by Himmelberg *et al.* [15]. To do so, we extracted the mean contrast thresholds across all simulated cone densities and mRGC:cone ratios. This resulted in a matrix of 17 cone densities x 5 mRGC:cone ratios. We placed these data points in a 3D coordinate space: log cone density (x-dimension) by log mRGC:cone ratio (y-dimension) by log contrast thresholds (z-dimension). We fitted a 3D mesh using a regression with locally weighted scatterplot smoothing with MATLAB's *fit.m* (using a LOWESS fit type with a span = 0.2, built-in normalization and the 'bisquare' robust fitting options). This 3D mesh fit is used to visualize the effect of cone density at a single mRGC:cone ratio by extracting a single curve from the mesh at that particular ratio (**Fig 6A**). We then used the 3D mesh fit to predict contrast thresholds for the four cardinal meridians at

4.5˚ eccentricity, evaluating the model at the four observed [cone, mRGC:cone ratio]-density coordinates reported by Curcio *et al.* [9] and Watson [64].

Predicted thresholds for the model up to cone isomerizations and photocurrent were computed using contrast thresholds for each cone density. These data were fitted separately per model stage, with the same 3D mesh fit as mRGC responses using a dummy variable for the mRGC:cone ratio. This fit was used to predict thresholds for each model stage given the observed cone densities at the four cardinal meridians at 4.5˚ eccentricity.

Contrast thresholds were converted into contrast sensitivity by taking the reciprocal. Nasal and temporal retina were averaged to represent the horizontal meridian. Because cone density can vary dramatically across observers [141,142], we computed error bars that represent the amount of variability in predicted sensitivity based on a difference in underlying cone density.

The upper/lower bound of the error bars in cone and mRGC model predictions were defined by assuming that our estimates of cone density on the meridians are imperfect. Specifically, we assumed that the measured asymmetries might be off by as much as a factor of 2. So, for example, if the reported density for the horizontal meridian is 20% above the mean, and for the vertical meridian is 20% below the mean, we considered the possibility that they were in fact 40% above or below the mean, or 10% above or below the mean.

## Supporting information

**S1 Fig. Polar angle asymmetries for cone density, mRGC density and V1 surface area computed from different publicly available datasets.** Asymmetries are in percent change, calculated as the difference between horizontal and vertical meridians divided by their mean (left column), the difference between upper and lower vertical meridians divided by their means (right column). Positive asymmetries would positively correlate with observed differences in behavior. (Top row) Cone data are from either Curcio *et al.* [9] (black lines) or Song *et al.* [10] (orange line) computed with either ISETBIO (solid lines) or rgcDisplacementMap toolbox (dotted lines). (Middle row) Midget RGC RF data are computed using the computational model by Watson (2014) implemented in the ISETBIO toolbox (solid black line) or Barnett and Aguirre [76] implemented in the rgcDisplacementMap toolbox (dotted black line). (Bottom row) V1 surface is computed from the Human Connectome Project 7T retinotopy dataset ($n = 163$), using the analyzed dataset by Benson *et al.* [78,79]. Surface areas are defined as ±10˚ (black) and ±20˚ (red) wedge ROIs from 1–6˚ eccentricity around the meridians, avoiding the central one degree and stimulus border (7–8˚) as those data can be noisy. Note that the x-axis is truncated as cortical measurements are limited by the field-of-view in the fMRI experiment. Data are fit with a 2nd degree polynomial, $R^2 = 0.48$ (±10˚) and $R^2 = 0.89$ (±20˚) for horizontal-vertical and $R^2 = 0.94$ (±10˚) and $R^2 = 0.72$ (±20˚) for vertical-meridian asymmetries).
(EPS)

**S2 Fig. Transformation ratios relative to horizontal visual field meridian.** Relative ratio is computed taking the lower or upper visual field transformation ratio and horizontal visual field transformation ratio from panel B, and divide the two for cone:mRGC ratios (left panel) and mRGC:V1 CMF ratios (right panel).
(EPS)

**S3 Fig. Classifier performance varying with cone density, separately for each mRGC:cone ratio.** Linear SVM classifier accuracy is computed for each contrast level in a simulated experiment with 1,000 clockwise and 1,000 counter-clockwise trials. Average accuracy data are fitted

with a Weibull function.
(EPS)

**S4 Fig. The effect of noise in mRGC layer on contrast thresholds as a function of cone density, separately for each mRGC:cone ratio.** (A) Contrast thresholds as a function of cone density when adding white noise following a Gaussian distribution with a standard deviation of 0.5 (left panel), 1 (middle panel), 2 (right panel). Data are fit with a locally weighted regression using the same procedure as the fit shown in Fig 6. Middle panel (1 std) is identical to Fig 6A. (B) Same data as panel A, visualizing the three mRGC noise levels separately per mRGC:cone ratio. Decreasing opacity of fits and data correspond to decreasing levels of noise.
(EPS)

## Acknowledgments

We thank Michael Landy and Brian Wandell for their useful comments.

## Author Contributions

**Conceptualization:** Marisa Carrasco, Jonathan Winawer.

**Formal analysis:** Eline R. Kupers, Noah C. Benson, Jonathan Winawer.

**Funding acquisition:** Marisa Carrasco, Jonathan Winawer.

**Investigation:** Eline R. Kupers, Noah C. Benson, Marisa Carrasco, Jonathan Winawer.

**Methodology:** Eline R. Kupers, Noah C. Benson, Jonathan Winawer.

**Resources:** Noah C. Benson.

**Software:** Eline R. Kupers, Noah C. Benson, Jonathan Winawer.

**Supervision:** Noah C. Benson, Marisa Carrasco, Jonathan Winawer.

**Visualization:** Eline R. Kupers, Jonathan Winawer.

**Writing – original draft:** Eline R. Kupers, Jonathan Winawer.

**Writing – review & editing:** Eline R. Kupers, Noah C. Benson, Marisa Carrasco, Jonathan Winawer.

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
