## [Decision Letter · Decision Letter 0]

15 Nov 2020

Dear Ms Kupers,

Thank you very much for submitting your manuscript "Radial asymmetries around the visual field: From retina to cortex to behavior" for consideration at PLOS Computational Biology.

As with all papers reviewed by the journal, your manuscript was reviewed by members of the editorial board and by several independent reviewers. In light of the reviews (below this email), we would like to invite the resubmission of a revised version that takes into account the reviewers' comments.

Sincerely,

Saad Jbabdi

Associate Editor

PLOS Computational Biology

Wolfgang Einhäuser

Deputy Editor

PLOS Computational Biology

Reviewer's Responses to Questions

**Comments to the Authors:**

Reviewer #1: Review uploaded as an attachment.

Reviewer #2: In “Radial asymmetries around the visual field: From retina to cortex to behavior,” the authors report a suite of insights into neural and perceptual asymmetries along the horizontal and vertical meridians of the retina. This is a thoughtful, detailed, creative, and extremely well written treatment of this issues, which makes several notable advances on prior work on this topic. The author’s recent prior work (also published in PLOS CB) focused on meridional asymmetries in optics and cone photoreceptor density; in the current report there are two key new insights. First, the authors systematically leverage anatomical and functional data to show that these asymmetries are present and amplified at the levels of retinal ganglion cell density and cortical magnification factor in early visual cortex. Second, they show that the spatial filtering properties of midget retinal ganglion cells, combined with asymmetric density long the meridians, augment the predicted asymmetries of observer models, but still do not account for the quite large differences measured in perceptual experiments. My comments that follow are all minor and mostly suggestions for ways to improve the clarity and completeness of the report.

Intro/Terminology: The authors use the term “radial asymmetries” to refer to variations as a function of polar angle (pg 3). I found this confusing. To me, these variations would be more accurately referred to as “polar angle asymmetries” or “meridional asymmetries.” The term, “radial asymmetries” implies that the variations appear as a function of radius. In pg 23, the authors even use the phrase “radial asymmetries” and “polar angle asymmetries” to refer to the same phenomenon of interest in two consecutive sentences. It doesn’t seem appropriate that both of these terms be used to refer to the same thing, and I think the latter is more clear.

Results:

-Pg. 6 (Fig 1): I didn’t understand why the mRGC : V1 CMF ratio decreases again after 20 degrees. Is this expected? It would be helpful if the authors addressed this feature of the data.

-Pg 8 (Fig 2): The CMF data derived from the HCP appear to have quite different mRGC : V1 CMF ratios than the ratios derived when the Horton & Hoyt formula is used (Fig 2B). Some of this difference might be because the HCP data are plotted only for the meridians. Would it be possible to add average lines for the HCP CMF data to Fig 2 panels A and B, for a more direct comparison to the formula overall? I also think the inclusion of V2 in the CMF estimates for the vertical meridian, but not the horizontal, warrants further scrutiny. It seems fair to compare superior and inferior in this regard, but is it really fair to compare horizontal to vertical with these data? In addition to the stated assumption that “V2 is approximately the same size as V1” (pg 29), do other assumptions need to be made about the relative CMF and receptive field sizes in V2 and V1?

-Pg. 12 (Fig 3): I understand that a spiking model isn’t used for the RGCs, but it doesn’t seem appropriate to label RGC responses in units of photons/ms. I’d suggest just calling these arbitrary units.

Pg. 14 (Fig 4): I found panel A confusing, possibly because the locations of the cones were not indicated. Overall, it would be helpful to have a visualization that includes the cone mosaic to show how the mRGCs tile it. Maybe it’s also hard to interpret this figure because the 2:1 ratio is spatially 1:1? (If one linear mRGC represents a pair of ON/OFF cells, pg 31).

Methods:

-Pg. 32: it should be noted that conv2 uses zero padding. It was unclear to me whether the sub-sampling removed any edge samples that include contributions of this padding. Is that the case? If not, does the padding impact the results?

-Pg. 35: I may be missing something, but I didn’t understand why interpolation was used here. Why not train classifiers directly on the appropriate ratios for each meridian at 4.5 deg?

General Formatting:

-In Fig 2, the different lines were hard to distinguish when printed. Given that the number of lines is pretty small, I’d suggest applying different line styles and/or saturations

-In the revised manuscript, it would be helpful to include line numbers in addition to page numbers

**Have all data underlying the figures and results presented in the manuscript been provided?**

Reviewer #1: Yes

Reviewer #2: None

PLOS authors have the option to publish the peer review history of their article (what does this mean?). If published, this will include your full peer review and any attached files.

Reviewer #1: No

Reviewer #2: No
---

## [Decision Letter · Decision Letter 1]

19 Dec 2021

Dear Ms Kupers,

We are pleased to inform you that your manuscript 'Asymmetries around the visual field: From retina to cortex to behavior' has been provisionally accepted for publication in PLOS Computational Biology.

Best regards,

Saad Jbabdi

Associate Editor

PLOS Computational Biology

Wolfgang Einhäuser

Deputy Editor

PLOS Computational Biology

Reviewer's Responses to Questions

**Comments to the Authors:**

Reviewer #1: The authors have addressed all my points/concerns and I have nothing more to add. Thank you for such a great paper !

Reviewer #2: No further comments.

**Have the authors made all data and (if applicable) computational code underlying the findings in their manuscript fully available?**

Reviewer #1: Yes

Reviewer #2: Yes

PLOS authors have the option to publish the peer review history of their article (what does this mean?). If published, this will include your full peer review and any attached files.

Reviewer #1: **Yes: **Nicolas Cottaris

Reviewer #2: No

---

## [Editor Report · Acceptance letter]

3 Jan 2022

PCOMPBIOL-D-20-01908R1 

Asymmetries around the visual field: From retina to cortex to behavior

Dear Dr Kupers,

I am pleased to inform you that your manuscript has been formally accepted for publication in PLOS Computational Biology. Your manuscript is now with our production department and you will be notified of the publication date in due course.

With kind regards,

Agnes Pap
